# Clinically SUspected ScaPhoid fracturE: treatment with supportive bandage or CasT? 'Study protocol of a multicenter randomized controlled trial' (SUSPECT study)

Abigael Cohen  ,[1] Max Reijman,[1] Gerald A Kraan,[2] Nina M C Mathijssen,[2] Marc A Koopmanschap,[3] Jan A N Verhaar,[1] Sander Mol,[4] Joost W Colaris,[1] On behalf of the SUSPECT study group

¹Department of Orthopaedics, Erasmus MC, Rotterdam, Zuid-Holland, The Netherlands
²Department of Orthopaedics, Reinier de Graaf Hospital, Delft, Zuid-Holland, The Netherlands
³Erasmus School of Health Policy and Management, Erasmus University Rotterdam, Rotterdam, Zuid-Holland, The Netherlands
⁴Department of Emergency Medicine, Franciscus Gasthuis en Vlietland, Rotterdam, Zuid-Holland, The Netherlands

**Correspondence to**
Abigael Cohen;
a.cohen.1@erasmusmc.nl

## ABSTRACT

**Introduction** Some scaphoid fractures become visible on radiographs weeks after a trauma which makes normal radiographs directly after trauma unreliable. Untreated scaphoid fractures can lead to scaphoid non-union progressing to osteoarthritis. Therefore, the general treatment for patients with a clinically suspected scaphoid fracture and normal initial radiographs is immobilisation with below-elbow cast for 2 weeks. However, most of these patients are treated unnecessarily because eventually less than 10% of them are diagnosed with an occult scaphoid fracture. To reduce overtreatment and costs as a result of unnecessary cast treatment in patients with a clinically suspected scaphoid fracture and normal initial radiographs, we designed a study to compare below-elbow cast treatment with supportive bandage treatment. We hypothesise that the functional outcome after 3 months is not inferior in patients treated with supportive bandage compared to patients treated with below-elbow cast, but with lower costs in the supportive bandage group.

**Methods and analysis** The SUSPECT study is an open-labelled multicentre randomised controlled trial with non-inferiority design. A total of 180 adult patients with a clinically suspected scaphoid fracture and normal initial radiographs are randomised between two groups: 3 days of supportive bandage or 2 weeks of below-elbow cast. We aim to evaluate the functional outcome and cost-effectiveness of both treatments. The primary outcome is the functional outcome after 3 months, assessed with the Quick Disability of the Arm, Shoulder and Hand score. Secondary outcomes include functional outcome, recovery of function, pain, patient satisfaction, quality of life and cost-effectiveness measured by medical consumption, absence from work or decreased productivity.

**Ethics and dissemination** The Medical Ethics Committee of the Erasmus MC Medical Centre, Rotterdam, approved the study protocol (MEC-2017-504). We plan to present the results after completion of the study at (inter)national conferences and publish in general peer-reviewed journals.

**Trial registration number** NL6976.

### Strengths and limitations of this study

► This is the first multicentre randomised controlled trial to evaluate if treatment with supportive bandage results in not inferior functional outcome to below-elbow cast in patients with a clinically suspected scaphoid fracture.
► All patients are re-evaluated clinically and radiographically after 1 year.
► The cost-effectiveness of the treatment up to 1 year is studied.
► The main limitation of the study is that the study is not blinded for participants, physicians and researchers.

## INTRODUCTION

Up to 32% of all scaphoid fractures result in non-union, leading to progressive osteo-arthritis.[1] While most fractures are visible on radiographs directly after trauma, not all scaphoid fractures can be diagnosed on the initial radiographs. To avoid untreated occult scaphoid fractures, all patients with a clinically suspected scaphoid fracture and normal initial radiographs are treated with a below-elbow cast until re-examination or further diagnostics.[2 3] Finally, less than 10% of these patients appear to have a scaphoid fracture resulting in a large number of overtreatment, unnecessary absenteeism from work and increased healthcare costs.[4 5]

Most occult scaphoid fractures can be diagnosed on repeated conventional radiography after 10–14 days.[6] These occult fractures are in general non-displaced fractures of the waist or distal pole of the scaphoid, in which an intact periosteal envelop provides stability.[7 8] Known risk factors for scaphoid non-union are proximal fracture location,

fracture displacement and start of treatment more than 4 weeks after trauma.[8–10] When adequate treatment for a scaphoid fracture is started within 4 weeks after the trauma, the non-union rate does not increase.[8] Therefore, supportive bandage and re-examination within 2 weeks can be a good alternative treatment for patients with a clinically suspected scaphoid fracture with normal radiographs. In 1988, one randomised study of 106 patients with a clinically suspected scaphoid fracture showed that treatment with supportive bandage compared with below-elbow cast in these patients decreased the immobilisation time and absence of work without negative consequences on fracture healing. However, functional outcome and cost-effectiveness were not evaluated, and the follow-up period was not clearly stated.[8]

We designed a pragmatic randomised controlled trial with 1-year follow-up to evaluate the functional outcome and cost-effectiveness of treatment with a supportive bandage compared with below-elbow cast in patients with a clinically suspected scaphoid fracture and normal initial radiographs. We hypothesise that treatment with supportive bandage results in a not inferior functional outcome compared with below-elbow cast, with lower costs in the supportive bandage group.

## Objectives

We designed a pragmatic, multicentre randomised controlled trial with a non-inferiority design with two groups. The primary objective of this study is to determine if treatment with supportive bandage compared with below-elbow cast in patients with clinically suspected scaphoid fracture and normal initial radiographs results in not inferior functional outcome after 3 months, measured with the Quick Disability of the Arm, Shoulder and Hand (QDASH).

As secondary objectives, we assess whether treatment of patients with a clinically suspected scaphoid fracture with a bandage is cost-effective compared with cast. Cost-effectiveness is evaluated by medical consumption, absence from work or decreased productivity, and quality of life, measured in quality-adjusted life years (QALYs).

Furthermore, we assess if supportive bandage compared with cast results in not inferior pain, patient satisfaction of the received treatment, functional outcome measured with the QDASH, recovery of function measured with physical examination and the Patient-Rated Wrist/Hand Evaluation (PRWHE) and quality of life.

## METHODS AND ANALYSIS

This manuscript is written according to the Consolidated Standards of Reporting Trials statement and the Standard Protocol Items: Recommendations for Interventional Trials guidelines.[11 12]

## Patient and public involvement

In the absence of a patient association, we formed a panel of three patients with a (suspected) scaphoid fracture to think along with, and comment on our study. This panel of patients analysed the study (aim, burden for patient, willingness to participate), patient information sheet and how the future results should be reported to the patients. To improve our study, we evaluate the study procedures and the patient burden with participating patients during the recruitment period. After termination of the study, we present our study results to all participating patients.

## Study design

The SUSPECT study is performed in nine hospitals in the Netherlands. All departments involved in the treatment of patients with a clinically suspected scaphoid fracture in each participating hospital participate in the study.

All patients who present with a clinically suspected scaphoid fracture after trauma at the emergency department (ED), who match the inclusion and exclusion criteria, are informed about the study by their treating physician. These patients receive the participant information sheet and are invited to ask questions about the study. As direct start of treatment is necessary, patients have to decide on study participation after the information is provided at the ED. Written informed consent is obtained from the patient prior to inclusion (online supplemental appendix 1). All included patients are randomised at the ED and allocated to treatment with supportive bandage or below-elbow cast.

Patients revisit the outpatient clinic after 2 weeks and 1 year. After 2 weeks, the researcher re-examines the patient first. Next, radiographs with scaphoid-specific views are obtained (without cast or bandage around the wrist) according to the hospital protocol. Last, the physician re-examines the patient and determines the diagnosis, treatment and follow-up according to their hospital protocol without interference of the researcher. When a scaphoid fracture is diagnosed by the physician, patients are treated for a scaphoid fracture according to hospital protocol with either cast or surgery. After 1 year, the researcher re-examines the patients and wrist radiographs are made.

During the follow-up period, patients receive questionnaires by email after inclusion, 2 weeks, 6 weeks, 3 months, 6 months, 9 months and 1 year of follow-up. (figure 1)

Due to the Dutch privacy law, it is not allowed to screen eligible patients at the ED to detect how many eligible patients are not participating in our study.

## Study population

All adult patients who visit the ED of the participating hospitals, within 48 hours after trauma, with a clinically suspected scaphoid fracture without a fracture on the radiograph, are invited to participate in the study.

## Inclusion criteria

► Aged 18 years or older.
► Trauma maximum of 48 hours before.
► Anatomical snuff box (ASB) tenderness or scaphoid tubercle (ST) tenderness.

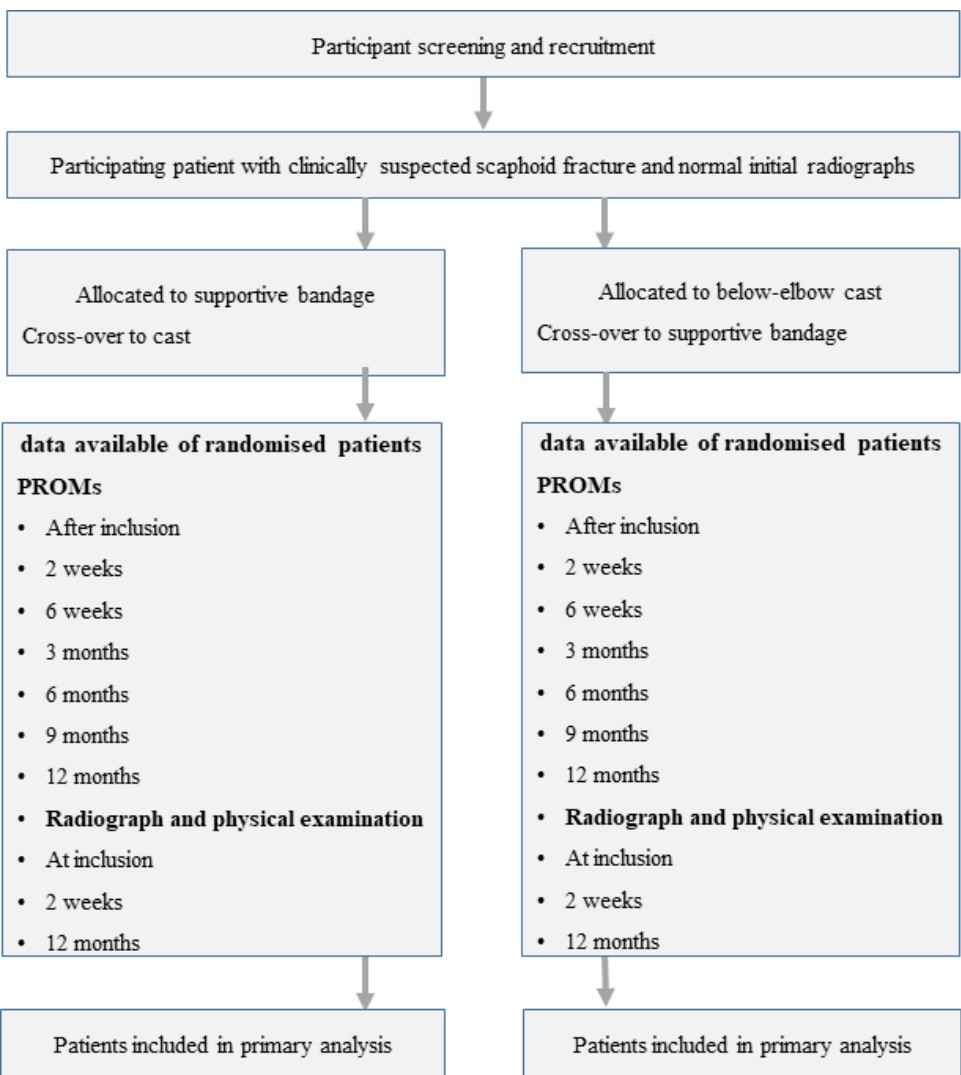

**Figure 1** SUSPECT study flow chart. PROMs, patient-rated outcome measures.

► No scaphoid fracture reported by the radiologist or treating physician on the initial radiographs with scaphoid-specific views (minimally three views).

### Exclusion criteria
► Concomitant injury of the ipsilateral extremity that needs treatment with movement restriction by cast or supportive bandage.
► Inability to complete study forms due to insufficient command of the Dutch language.
► A scaphoid fracture diagnosed on the initial radiographs by a supervising radiologist in retrospect.

### Randomisation
After written informed consent is obtained, patients are centrally randomised through computer-based variable block randomisation (2, 4, 6 blocks) by Castor Electronic Data Capture (EDC) to enable allocation concealment.[13] The allocation ratio is 1:1 and randomisation is stratified per hospital due to potential differences in applied cast and the number of scaphoid-specific views per hospital .

Because of practical reasons, there is no blinding of the treatment group for the physician, patient and researcher. We do perform the assessment of radiographs and statistical analysis both blinded.

### Interventions
#### Intervention group (supportive bandage)
Patients in the intervention group receive a below-elbow supportive bandage for 3 days at the ED. After 3 days the patients are allowed to remove the bandage and move the wrist driven by the amount of pain. When patients experience too much pain despite adequate analgesia, an additional appointment is made at the outpatient clinic to apply a below-elbow cast for analgesic purposes. The applied cast is similar to the control group, conform hospital protocol. The patients who cross over to cast remain in the study.

#### Control group (below-elbow cast)
The usual care in the Netherlands for patients with clinically suspected scaphoid fracture at the ED is below-elbow

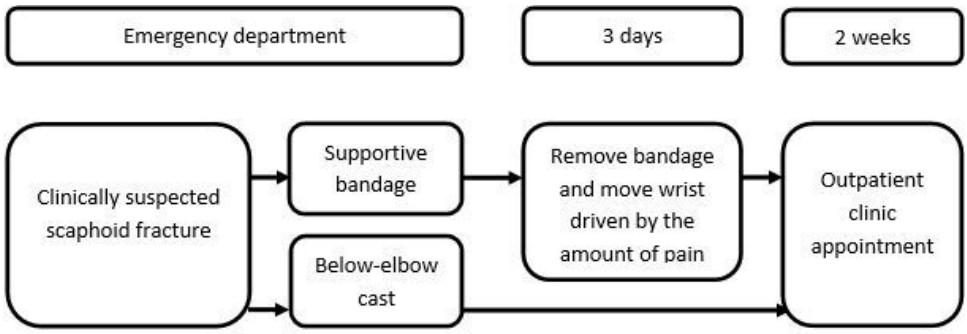

**Figure 2** Treatment of intervention and control groups.

cast and re-examination after 2 weeks (figure 2).[2] The applied below-elbow cast type is according to hospital protocol and can be circular or splint, with or without the thumb included.

### Outcome
We collect the following baseline variables: demographic data (eg, age, gender, height, weight, hand dominance), details about the trauma (eg, fall onto an outstretched hand, high-energy trauma, low-energy trauma, sports injury) and comorbidities such as smoking status, previous arm injuries and general medical history.

### Primary outcome
The primary outcome is the difference in functional outcome between the intervention group and control group measured with the QDASH after 3 months of follow-up. We selected the QDASH as the primary outcome, as it takes the function of both hands into account. With this information, we can reflect if the wrist problem results in an impaired functional outcome in daily life.

### Secondary outcomes
- ► Functional outcome measured with the QDASH after inclusion and at 2 weeks, 6 weeks and 1 year of follow-up (table 1).
- ► Recovery of function evaluated with PRWHE after inclusion and at 2 weeks, 6 weeks, 3 months and 1 year of follow-up.
- ► The amount of pain scored with the visual analogue scale (VAS) at rest and during movement after inclusion and at 2 weeks, 6 weeks, 3 months and 1 year of follow-up.
- ► Patient satisfaction of the treatment is measured after 2 weeks and 3 months of follow-up.
- ► Quality of life assessed by the 5-level EuroQol-5D (EQ-5D-5L) after inclusion and at 2 weeks, 6 weeks, 3 months and 1 year of follow-up. When the patient reports medical consumption or decreased productivity due to the wrist problems after 6 or 9 months of follow-up, the EQ-5D-5L is sent to the patients as well.
- ► To assess the costs and cost-effectiveness of both interventions, we collect information about medical consumption (eg, additional imaging and therapeutic hospital procedures) and productivity costs We use iMTA Medical Consumption Questionnaire (iMCQ) and iMTA Productivity Cost Questionnaire (iPCQ) after 6 weeks, 3 months, 6 months, 9 months and 1

**Table 1** Description of patient-rated outcome measures and examinations per follow-up moment

|  | After Inclusion | 2 weeks | 6 weeks | 3 months | 6 months | 9 months | 1 year |
|---|---|---|---|---|---|---|---|
| Radiograph | Scaphoid | Scaphoid |  |  |  |  | Wrist |
| Physical examination | x | x |  |  |  |  | x |
| PROM | QDASH PRWHE VAS EQ-5D-5L | QDASH PRWHE VAS Patient satisfaction EQ-5D-5L | QDASH PRWHE VAS EQ-5D-5L iMCQ iPCQ | QDASH PRWHE VAS Patient satisfaction EQ-5D-5L iMCQ iPCQ | iMCQ iPCQ (EQ-5D-5L) | iMCQ iPCQ (EQ-5D-5L) | QDASH PRWHE VAS EQ-5D-5L iMCQ iPCQ |

EQ-5D-5L, 5-level EuroQol-5D; iMCQ, iMTA Medical Consumption Questionnaire; iPCQ, iMTA Productivity Cost Questionnaire; PROM, patient-rated outcome measure; PRWHE, Patient-Rated Wrist/Hand Evaluation; QDASH, Quick Disability of the Arm, Shoulder and Hand; VAS, Visual Analogue Scale Pain at rest and during movement.

year of follow-up (table 1) and we screen the electronic patient status.

► Physical examination is performed at inclusion, after 2 weeks and 1 year of follow-up. At inclusion, the physical examination is performed by the treating physician. They register if the patient has tenderness ASB, tenderness ST, compression pain thumb, radial sided wrist pain with resisted supination and radial sided wrist pain with ulnar deviation. A previous study showed that these clinical tests have the highest sensitivity for a scaphoid fracture.[14] After 2 weeks and 1 year, the function of the wrist is assessed by the researcher. The physical examination includes the above-mentioned clinical tests and swelling or haematoma on the radial side of the wrist. Range of motion of the wrist (palmar flexion, dorsal flexion, ulnar deviation, radial deviation, supination and pronation) and finger-to-palm distance are measured with a goniometer. The finger-to-palm distance is the distance from the tip of the finger to the distal palmar crease, when the fingers are in maximal active flexion.[15] We assess the opposition of the thumb with the Kapandji score.[16] The Kapandji score ranges from 0 to 10, where 0 indicates no opposition and 10 maximal opposition. Grip strength is measured with a hand dynamometer (Jamar hand dynamometer), and is defined as the maximum grip strength after three attempts. Range of motion and grip strength of both wrists are examined. One year after inclusion, we additionally test the scapholunate (SL) ligament by identifying tenderness over the dorsal SL interval, performing the finger extension test and the Watson

### Table 2 Physical examination per follow-up moment

| | At Inclusion | 2 weeks | 1 year |
|---|---|---|---|
| Swelling radial side wrist | | x | x |
| Haematoma radial side wrist | | x | x |
| Tenderness ASB | x | x | x |
| Tenderness ST | x | x | x |
| Compression pain thumb | x | x | x |
| Painful resisted supination | x | x | x |
| Painful UD | x | x | x |
| SL ligament | | | x |
| Kapandji score | | x | x |
| Range of motion | | x | x |
| Finger-to-palm distance | | x | x |
| Grip strength | | x | x |

Range of motion includes palmar flexion, dorsal flexion, ulnar deviation, radial deviation, supination and pronation.
ASB, anatomical snuff box; SL, scapholunate ligament examinations which consist of tenderness SL interval, finger extension test and Watson test; ST, scaphoid tubercle; UD, ulnar deviation.

test (table 2). The finger extension test is positive when resisted extension of the fingers with the wrist in flexion is painful.[17] To perform the Watson test, the patients' wrist is in slight dorsal flexion and ulnar deviation, with the researchers' thumb on ST and fingers wrapped around the distal radius. Then, the researcher moves the patients' wrist radially and in palmar flexion. The Watson test is positive if a painful clunk is noticed.[17 18]

► Radiographic re-examination is performed after 2 weeks and 1 year of follow-up. After 2 weeks, the radiographs include scaphoid-specific views (minimally three views) to identify a scaphoid fracture. After 1 year, a posterior-anterior and lateral wrist radiographs are made to determine the presence of any post-traumatic injuries (eg, scaphoid non-union, SL dissociation). The radiographs are evaluated by two independent assessors.

► Detailed information about the received treatment is reported. We address the number of patients who cross over from their allocated treatment, days of cast or supportive bandage before the outpatient department appointment and applied treatment after the outpatient department appointment (cast or surgery).

► (Serious) adverse events are noted, such as scaphoid fracture and scaphoid non-union. (Detailed information can be found at safety considerations.)

► Other diagnoses are reported such as other carpal fractures or ligament injuries.

### Instruments

Functional outcome is measured with the mandatory module of the QDASH. The QDASH is the short version of the DASH and consists of 11 questions about symptoms and physical function of both arms during the last week. Each question can be scored from 1 to 5. The total score ranges from 0 - 100 (high score indicates severe disability of the upper extremity) The Dutch QDASH has shown good validity, reliability and responsiveness for comparable patients.[19 20]

The PRWHE questionnaire evaluates the painful wrist during the last week. The questionnaire consists of three subscales about pain, function and cosmetics. The cosmetic field is not included in the score. The pain and function subscales consist each of 15 questions that can be scored from 0 to 10. The total PRWHE score ranges from 0 - 100 (high score indicates severe pain and impaired function of the painful wrist). The Dutch PRWHE has shown good validity, reliability and responsiveness for comparable patients.[20]

Pain scores at rest and during movement are examined with the VAS for pain with a range from 0 - 10. A high VAS score indicates severe pain. The VAS for pain is widely used in research with comparable patients.[21]

Patient satisfaction is assessed by the questions 'how satisfied are you with the treatment', 'would you prefer the other treatment at the ED under similar circumstances?'

and 'if yes, why?'. The first question can be scored on a VAS scale 0 to 10. A high score indicates satisfaction.

Quality of life is measured with the EQ-5D-5L. This questionnaire consists of five questions and a VAS. The questions are about mobility, self-care, usual activities, pain or discomfort and anxiety or depression. Each question can be scored as: no problem, slight problem, moderate problem, severe problem and extreme problem. The index score ranges from 0 (death) to 1 (perfect health). Health is reported on a VAS from 0 to 100. A high score indicates better health state. The EQ-5D-5L is recommended for the assessment of quality of life in trauma patients, especially for economic assessments.[22]

To assess the costs and cost-effectiveness of the intervention with the usual care we include the Medical Consumption Questionnaire (iMCQ) and Productivity Cost Questionnaire (iPCQ). The iMCQ measures the total medical consumption by measuring the use of diagnostics, consultations, ED visits, physical therapy, medication and aids. The iPCQ measures productivity costs. This questionnaire evaluates costs of absence of work and decreased productivity at paid or unpaid work. The iMCQ and iPCQ are short generic measurement instruments.[23]

## Sample size

Our hypothesis is that patients with a clinically suspected scaphoid fracture with normal initial radiographs treated with a supportive bandage have no inferior functional outcome after 3 months compared with below-elbow cast, but with lower costs. The power calculation is based on the proof of non-inferiority. We took 50% of the margin of clinical minimal important difference of QDASH.[24] This resulted in a margin of non-inferiority of 7.50 points on the QDASH. The SD of QDASH after 3 months has been reported to be 14.[25] We used a power of 90% and a one-sided alpha of 0.025. To detect non-inferiority of supportive bandage compared with below-elbow cast 148 patients are needed (74 patients per group). Accounting for a 15% lost to follow-up, a total of 180 patients are required. When the lost to follow-up or change of allocated treatment is higher than 15%, we will include more patients to have 74 patients per group to answer our primary research question. When the patient recruitment rate at the ED is behind our expectations, we invite more hospitals to participate.

## Data collection

We collect data by electronic data capture with Castor EDC.[13] Paper-based case report forms are used at the ED and the outpatient department. The researcher inserts these data in Castor EDC. Questionnaires are sent directly to the patients through email via Castor EDC. In case an email address is not available, a paper case report form is sent to the patient. When the patient does not respond, we will contact the patient by telephone.

We screen the electronic patient status to collect information about (serious) adverse events, hospital visits, additional imaging and treatment for the wrist injury during the follow-up period of the study.

## Data analysis

We perform all analyses blinded for treatment allocation. The distribution analysis of baseline variables is tested by the Shapiro-Wilk test. For normally distributed variables, parametric tests are used. For those variables that are not normally distributed, non-parametric tests are used. We report the proportion of diagnosis and adverse events per allocated treatment group.

All outcomes are analysed 'by intention to treat analysis' with a linear mixed model with specified fixed and random effects. The covariance structure is unstructured. Restricted maximum likelihood will be used to estimate parameters. The assumptions of our model (linearity, homoscedasticity and normal distribution of the error terms) are tested. Should any of these assumptions seriously fail, variable transformations are used.

For our primary outcome, QDASH at 3 months, and secondary outcomes PRWHE, VAS score, EQ-5D-5L measurement time is a fixed effect and we add an interaction term of measurement time by treatment. Hospital is a random intercept and to analyse repeated measurements, we add a random intercept for patients (nested with hospital). The data after inclusion, and after 2 weeks, 6 weeks and 3 months of follow-up are included in our model as repeated measures. We perform post hoc analyses to compare estimated means at 3 months. All outcomes are assessed for non-inferiority, expect cost-effectiveness which is assessed for superiority. A margin of non-inferiority of 7.50 is used for our primary outcome in the analyses. Since our groups are small, it can be that the baseline characteristics are not in balance between the two groups. Therefore, we add the potential confounders as age, gender and presence of comorbidities that influence the function of the arms, as fixed effects. We perform additional as-treated analysis to analyse outcomes between patients who received only supportive bandage until the outpatient department appointment after 2 weeks and patients who received cast in the first 2 weeks (received cast at the ED or crossover from bandage). Patients are included in our mixed models when they responded to the outcome questionnaire irrespective of the number of time points. If no data of the outcome questionnaire are available, we report the reasons in detail in our paper and discuss the potential bias. When one or more (but not all) of the measurements are missing, a sensitivity analysis is performed to clarify if the missing data are at random.

Furthermore, we assess the progress of the QDASH, PRWHE, VAS, EQ-5D-5L and physical examination during the 1-year follow-up. The random and fixed effects remain similar, only all follow-up moments are included in the repeated measurements.

## Cost-effectiveness analysis

### General considerations

We assess the cost-effectiveness of 3 days of supportive bandage versus 2 weeks of below-elbow cast over a period of 1 year. To answer this question, we perform a cost-utility analysis to estimate the incremental cost per QALY gained, to be able to compare our study with other studies in musculoskeletal disorders and in other disease categories. We impute missing data by multiple imputation, using the pattern mixture model. Using non-parametric bootstrapping (randomly drawing 5000 observations with replacement from the patient sample), the degree of uncertainty for costs and health effects and the cost-utility ratio are depicted in a cost-effectiveness plane. In addition, an acceptability curve is drawn, which indicates the probability that the intervention studied has lower incremental costs per QALY gained than various thresholds (dependent on disease severity) for the maximum willingness to pay for an extra QALY.

### Cost analysis

The economic analysis is based on the societal perspective, and on the healthcare perspective in which the direct medical and productivity costs between the groups are compared. To measure the direct costs, an inventory of patients' total medical consumption is made using the electronic patient status and the iMCQ. The use of diagnostics (additional radiographs, CT, MRI, ultrasound or bone scintigraphy), consultations, ED visits, physical therapy, medication and aids are evaluated. Productivity costs are measured with the iPCQ. Costs of absence of work and decreased productivity at a paid job or at unpaid work are evaluated. The friction cost method is used to calculate the productivity costs according to the recent Dutch guidelines. The costs per unit of medical consumption are estimated, using information from the most recent Dutch Manual for economic evaluation of healthcare. Costs are reported for the year 2019.

### Cost-utility analysis

Patient outcomes relevant for the cost-effectiveness analysis are the number of QALYs during 1 year. Patient scores on the EQ-5D-5L version are converted into utility values, using the valuation algorithm of the Dutch general population.

### Budget impact analysis

Guidelines of the ISPOR Task Force are used for the budget impact analysis (BIA). Relevant features and tariffs of the Dutch healthcare system, anticipated uptake of the new intervention as well as usual care are taken into account. The size and the eligibility of the population, cost of diagnostics, treatment modalities and changes expected in condition-related costs are considered in the BIA. Sensitivity analyses are performed using divergent scenarios from the viewpoint of the decision-makers.

The budget impact per year of implementing the new intervention is estimated. All elements of medical costs for the new intervention as well as for usual care that are paid by third-party payers are considered and calculated.

## ETHICS AND DISSEMINATION

### Safety considerations

The study compares treatment of below-elbow cast (standard treatment) with supportive bandage (intervention). Since the study is labelled as low risk, a data safety monitoring board is not required. The study is monitored at least once a year and the progress of the trial is reported once a year to the accredited Medical Review Ethics Committee (MEC). Adverse events are recorded by the researcher during follow-up. A serious adverse event is reported (www.toetsingonline.nl) within a maximum of 15 days after the researcher is notified.

Adverse events are defined as:
- Cast-related problems.
- Bandage-related problems.
- Loss of function.
- Complex regional pain syndrome.
- Persisting pain after 6 weeks.

Serious adverse events related to our study are defined as:
- Surgically treated scaphoid fracture.
- Compartment syndrome.
- Scaphoid non-union (scaphoid fracture without signs of healing after minimally 12 weeks).

Because of the multicentre study design, clinical trial site agreements are obtained between the initiating and the participating hospitals. These agreements include liability and insurance aspects. The patient information sheet reports on participants' insurance and potential complications as a result of participation in the study. Withdrawal from the study is always possible for included patients without any consequences.

### Ethics

The study is approved by the MEC of the Erasmus Medical Centrum, Rotterdam, the Netherlands (MEC-2017-504). All amendments are notified to the MEC and changes to the study are made after a favourable opinion of the MEC.

The study is conducted according to the principles of the Declaration of Helsinki (64th version, date: October 2013) and in accordance with the Medical Research Involving Human Subjects Act (WMO) and other guidelines, regulations and acts.

The study is registered in the Dutch Clinical Trials Registry (www.trialregister.nl). Participating patients and physicians are not paid for participation in the study.

### Data management

Personal data of the participating patients that we collect during the study are changed to a study number. This study number is used for all study documentation, study reports and publications.

Data are collected with Castor EDC. Paper case report forms are entered in Castor EDC by the researcher. The

paper case report forms are securely filed in the hospital. All data are stored for 15 years. The final trial data set is accessible for the research team.

## Dissemination

After completion of the study, we plan to present the results at (inter)national congresses and submit the manuscript to general peer-review journals.

Authorship is according to the International Committee of Medical Journal Editors guidelines and a SUSPECT study group is made.

## Trial status

We are currently recruiting patients for the study while submitting the manuscript (14-01-2020). The date of first enrolment was 7 June 2018.

**Collaborators** SUSPECT study group: Peer van der Zwaal (Department of Orthopaedic Surgery, Haaglanden Medical Centre, The Hague, The Netherlands), Merel van Loon (Department of Emergency Medicine, Haaglanden Medical Centre, The Hague, The Netherlands), Steven J Rhemrev (Department of Surgery, Haaglanden Medical Centre, The Hague, The Netherlands), Daphne A van Rijssel (Department of Emergency Medicine, Reinier de Graaf Hospital, Delft, The Netherlands), Mark R de Vries (Department of Surgery, Reinier de Graaf Hospital, Delft, The Netherlands), Flip van Beek (Department of Surgery, Franciscus Hospital, Rotterdam, The Netherlands), Vanessa Brown (Department of Emergency Medicine, Franciscus Hospital, Rotterdam, The Netherlands), Tjebbe Hagenaars (Trauma Research Unit, Department of Surgery, Erasmus MC University Medical Center, Rotterdam, The Netherlands), Els H Jansen (Department of Emergency Medicine, Erasmus MC University Medical Center, Rotterdam, The Netherlands), Alexander P A Greeven (Department of Surgery, Haga Teaching Hospital, The Hague, The Netherlands), Lenneke C M Scholtens (Department of Emergency Medicine, Haga Teaching Hospital, The Hague, The Netherlands), Ruud L M Deijkers (Department of Orthopaedic Surgery, Haga Teaching Hospital, The Hague, The Netherlands), Niels W L Schep (Department of Surgery, Maasstad Hospital, Rotterdam, The Netherlands), Akkie N Ringburg (Department of Surgery, Ikazia Hospital, Rotterdam, The Netherlands), Milko M M Bruijninckx (Department of Surgery, Ijsselland Hospital, Capelle aan den Ijssel, The Netherlands), Kirsten F van Meerten (Department of Emergency Medicine, Albert Schweitzer Hospital, Dordrecht, The Netherlands), Willem-Maarten P F Bosman (Department of Surgery, Albert Schweitzer Hospital, Dordrecht, The Netherlands).

**Contributors** MR and JWC conceived the study idea and secured funding. AC, MR, GAK, NMCM, MAK, JANV, SM and JWC were involved with the initial study design. MR and MAK were involved with the statistical analysis for the study. AC, GAK, SM, JWC and SUSPECT study group are involved with the implementation of the protocol and the data collection. AC conducted the study. All authors provided critical comments to the manuscript and approved the final version before submitting.

**Funding** For this study, we received a grant from ZonMw, a Dutch organisation for health research and care innovation, and cofunding from CZ, a Dutch health insurance company.

**Competing interests** None declared.

**Patient consent for publication** Not required.

**Provenance and peer review** Not commissioned; externally peer reviewed.

**ORCID iD**
Abigael Cohen http://orcid.org/0000-0001-9412-1134

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
