## [Reviewer comments · BMJ Open]

ARTICLE DETAILS

TITLE (PROVISIONAL)	clinically SUSpected ScaPhoid fracture: treatment with supportive bandage or CasT? "Study protocol of a multicenter randomized controlled trial" (SUSPECT study)
AUTHORS	Cohen, Abigail; Reijman, M; Kraan, Gerald; Mathijssen, Nina; Koopmanschap, Marc; Verhaar, Jan; Mol, Sander; Colaris, Joost

VERSION 1 – REVIEW

REVIEWER	Christina Brudvik Department of Clinical Medicine University of Bergen Norway
REVIEW RETURNED	18-Feb-2020

GENERAL COMMENTS	Review comments to the SUSPECT-study: This study is important and interesting because it questions the foundation of an established "truth" about how to treat clinically suspected scaphoid fractures, with initially normal radiographs. The fear of not immobilizing a scaphoid fracture early enough and run the risk of non-union or avascular necrosis, is most likely exaggerated. The researchers assume that most scaphoid fractures with initially normal x-rays are less likely to heal with complications. The study design is well founded, but the following points should be addressed: The researchers ought to be blinded for which sort of immobilisation the patient has had, when they perform their extensive examination after 2 weeks. This would make the results more trustworthy, and it is not enough to just let the radiologists be blinded. This can be done if new X-rays are taken before the researcher examines the patient. The radiographers can remove casts and bandages and make patients wash their hands prior to this examination. The planned extensive examination by the researcher, 2 weeks after injury, might not be possible for patients to perform, and might even be harmful in case of a scaphoid fracture (eg measuring the hand strength with a Jamar dynamometer). What happens with patients with too much pain to follow the short bandage randomisation group after some time? Are they excluded from the study? Is it necessary for the researcher to do the clinical examination after 2 weeks at all - unless the researchers intend to validate the physicians' clinical skills? The study would have a higher impact if
--

	the patient-flow is done as normal and close to the every-day clinical life as possible. If this physical examination after 2 weeks is done by the physician on call - as in normal practice, the researcher would be blinded for the examination after 1 year. The assessments of the different PROM-results should also be done in a blinded way. Is it necessary to use so many different PROMS and so many follow up-periods for patients to answer the PROMS? The drop-out risk will be high because of this. Some PROMS address the same factors. Quick-Dash and PRWHE should be preferred and both already include a pain scale. Another major concern is the number of patients included. The sample size is based on significant differences in PROM scores. Even if the research question is “whether clinically suspected scaphoid fractures without radiological evidence of a fracture – have an equal outcome after 3 months if they use a bandage or a cast”, it is of some importance to address whether they have an identified scaphoid fracture or not after 2 weeks. With only 180 patients included and 10% with a scaphoid fracture (n=18), the risk of large baseline differences in the randomization is present. The risk of type 1 statistical errors is high if eg most of the fractures are immobilised in a cast. Likewise, age and gender should be equally distributed as these factors affect both pain scores and function. Should there be an upper age limit - like 50 years of age? Higher risk of radial sided pain due to CMC-osteoarthritis? A discussion about all the other possible types of injury in a “radial sided painful wrist after injury” is missing in the introduction. Scapholunate ligament-injuries, painful bone-bruises, etc. It is not discussed whether it is clinically relevant to do an X-ray of all wrists after 2 weeks. Some patients have no pain and no clinical signs at all, and the medical indication for a new X-ray might not be present. Sometimes an MRI or CT scan is needed after two weeks in order to identify a scaphoid fracture that is still not evident on X-rays, but still clinically suspected. These investigations are expensive and will have an impact on the cost analysis. This should also be addressed. A more detailed and complete flow-chart of the study is needed.
--	---

REVIEWER	Stephen Derek Brealey University of York, United Kingdom
REVIEW RETURNED	22-Feb-2020

GENERAL COMMENTS	This is an interesting study and I enjoyed reading the protocol. I have some comments below. In the “strengths and limitation of this study” it comments that the study is looking at supportive bandage producing equal function outcome to below-elbow cast. The design, however, is not looking at whether the two treatment options produce equal outcomes but whether the bandage is “not unacceptably worse” than cast. The authors also state at the end of the “Introduction” that their hypothesis is that the two treatments will be equal in functional outcome. This is not the hypothesis that is being tested with the choice of trial design. Again, in the “sample size” section, it comments that looking at “minimally equal function”. I think the choice of language about being “equal” is misleading. You are willing to accept introducing into clinical practice an inferior
--

	treatment, as long as its not unacceptably worse and because of its potential cost-effectiveness. I think the design is fine, with the minimal risk to patients, but it is not producing findings that treatments are “equal”. If <10% of patients have an occult scaphoid fracture and are concerned about over-treatment and costs, then should there be a third arm of ‘no treatment’? The occult fractures are also likely to be the lesser risk fractures to lead to non-union and, therefore, further reduces the risk of ‘no treatment’. The two-arm design is acceptable, but was a ‘no treatment’ arm considered with two week radiological follow-up and patients treated timely in a cast to prevent non-union? I understand though that it could be argued that a patient needs a little support and a bandage is low cost. It should be clear in the “Study design” section what imaging (X-ray series and/or CT) the patient gets on first presentation to the hospital. The standard of imaging taken on initial presentation can help reduce missing no fracture. Even if its according to local hospital protocol, rather than an agreed trial protocol, this should be stipulated. I can understand in a pragmatic trial you may not want to be too prescriptive about what imaging is taken, but I would prefer to see at least a minimal standard of X-rays taken across the hospitals – particularly when trying not to miss a difficult fracture to spot. Also, only two radiographs at one year (PA and lateral) to assess for non-union? This is a very limited set of imaging – if you are to bring in the patient at one year for extra little cost and radiation exposure you could collect a more thorough set of imaging and hence rigorous assessment for non-union. As would including more than one person doing this assessment. How is “non-union” defined? Also, in the “Study design” section, patients will be asked to complete questionnaires at baseline and at six further follow-ups over a year. This is considerable (arguably unnecessary) patient burden, for whom over 90% won’t even have a fracture. Most patients at three months will have largely recovered from their fracture, even if they have one. I would have advised against the six and nine month questionnaires, particularly when the three month questionnaire is the primary end-point, and the one year as a final check (although this final check could have been done at six months as one year on will be difficult to bring patients in for review and collect further data with no financial incentives/expenses to be covered). Be explicit in the “Intervention group” about how the below-elbow cast will be applied like it is for the “control group”. Also, how did you arrive at “3 days”? You could have a lot of patients coming in for cast because still in pain after only 3 days – this could lead to a lot of cross-over? I am not sure what the protocol is when a fracture is found at the two week review. When a patient does have a fracture and its only being treated in cast how are they still followed-up to monitor for non-union? If non-union does occur will surgery be performed? If in the bandage group and there is a fracture after the two week review, do you switch to cast or keep with the bandage? In the “sample size” section it is specified that the margin of non-inferiority is 7.50 points on the QDASH. The standard deviation
--	--

	after three months is expected to be 14 points. This equates to a large effect size of 0.53. This appears to mean that you are willing to accept the bandage being up to half a standard deviation inferior on the QDASH. Is this an acceptable amount of inferiority?
--	--

REVIEWER	Suzie Cro Imperial Clinical Trials Unit, Imperial College London, UK
-----------------	---

REVIEW RETURNED	03-May-2020
-------------

GENERAL COMMENTS	Line 132-135 Objectives: the authors state “The primary objective of this study is to determine if treatment with supportive bandage compared with below-elbow cast in patients with clinically suspected scaphoid fracture and normal initial radiographs, results in equal functional outcome after 3 months” – however the authors are conducting a non-inferiority trial, not an equivalence trial. Can the authors clarify here that the objective is to test that supportive bandage is not inferior to below elbow cast– rather than testing for equivalence, which i assume it is given the design. Line 138-140 “we assess post-traumatic injuries, pain, patient satisfaction of the received treatment, functional outcome measured with the QDASH, recovery of function measured with physical examination and the Patient Rated Wrist/Hand 140 Evaluation (PRWHE) and quality of life ”Are secondary outcomes being assessed for non-inferiority (as the primary objective) or will be superiority be assessed for these? Please further clarify the secondary objectives. Line 149 – Patient and public involvement statement – Great that the authors has a patient panel. Can the authors add how many patients made up the patient panel? Line 197 Randomization – presumably the computer-based randomization enables allocation concealment i.e. although an open trial those involved in recruitment are not aware of the next randomisation? Can the authors explicitly state this if this is the case for transparency on allocation concealment. Sample size – line 325 – can the authors clarify the primary hypothesis of the study. In a non-inferiority trial, the null hypothesis is that the new treatment is inferior to the standard control treatment. The alternative research hypothesis is that the experimental treatment is not inferior to the standard control arm by the pre-specified margin. Data analysis: Line 354 - It is written that “Adjustment are done for those baseline variables that change the effect estimate with more than 10%” This is an usual approach to take. Can the authors provide the list of potential prognostic variables that will be assessed using this approach? it is recommend that only variables known a priori to be strongly/moderately related with the primary outcome be considered as covariates and pre-specified In the protocol and that only a few justified covariates should be included in a primary analysis. (see https://www.ema.europa.eu/en/documents/scientific-guideline/guideline-adjustment-baseline-covariates-clinical-trials_en.pdf which states “Known or expected associations with the primary outcome variable should be justified on the basis of previous evidence (e.g. data from previous trials) and/or on clinical
--

	grounds. The reasons for including a covariate in the primary analysis should be explicitly stated in the protocol.”) Will the baseline outcome be adjusted for within the analysis? This should usually be included as a covariate. (see https://www.ema.europa.eu/en/documents/scientific-guideline/guideline-adjustment-baseline-covariates-clinical-trials_en.pdf) Given randomisation is stratified by site I was surprised that it is not planned to adjust for site in the analysis (see above reference and Kahan, B.C. and Morris, T.P. (2013), Analysis of multicentre trials with continuous outcomes: when and how should we account for centre effects?. Statist. Med., 32: 1136-1149. doi:10.1002/sim.5667 and Kahan, B.C. and Morris, T.P. (2012), Improper analysis of trials randomised using stratified blocks or minimisation. Statist. Med., 31: 328-340. doi:10.1002/sim.4431). It is recommend that analysis should adjust for recruitment centre to obtain the appropriate standard errors. No details on provided how missing data will be handled. Given up to 15% anticipated in sample size calculations please provide details on how missing data will be handled in the primary analysis. Are any sensitivity analyses planned if there is up to 15% missing data? methods for sensitivity analysis should be included where planned. Line 356-60: Secondary scores will be analysed with a linear mixed models. The authors list the fixed effects – presumably treatment should also be included here? Will the trialists be assessing the treatment effect over all time periods, i.e. analysis model will include just a single fixed treatment effect? Or will the model include a treatment by time interaction to assess the treatment effect at each time point separately? Can the authors indicate whether there will be any random effects in the mixed model? And if so which ones? Are secondary outcomes being assessed for non-inferiority or will a superiority/other comparison be assessed for these? How will adverse events be described/analysed? and for what analysis population? Please provide details. Cost effectiveness analysis - please provide details on how missing data will be handled within these analysis. Can the authors include a brief update on trial status/dates, e.g.. On what date did recruitment begin and when is data for last recruitment/follow-up anticipated?
--	--

REVIEWER	Aleksandra Turkiewicz Clinical Epidemiology Unit, Orthopedics, Clinical Sciences Lund, Lund University
REVIEW RETURNED	12-May-2020

GENERAL COMMENTS	Statistical comments: The protocol describes details of a randomized multicentre study to evaluate if supportive bandage is non-inferior to below-elbow cast in patients with a clinically suspected scaphoid fracture. The
--

	primary outcome is QDASH measured at 3 months after randomization and the study is performed in 9 hospitals in Netherlands. The planned non-inferiority margin is half of clinically important difference. Study design.  1. The study assumes block randomization stratified by centre with varying block size, with allocation ratio 1:1. Could you please clarify how the randomization is done in practice? Does the treating physician obtain the treatment allocation through the hospital computer system? Or will numbered sealed envelopes be used? 2. Will the proportion of patients required below-elbow cast at the 14 days visit be reported? If it would theoretically be that most of the patients still require cast (even with equal functional results at 3 months) that would suggest a potential need not to delay the cast treatment. Could you please comment on this? 3. It would be preferable if the examining physician for 1-year outcome was not aware of the treatment received. Is this feasible? 4. Even if blinding of treating physicians is not possible, it should be possible for the investigator and the statistician to be blinded. Please clarify if this is planned or not and why. 5. The study design includes a possibility of cross-over from supportive bandage group to the cast group, when cast is needed for analgesic purposes after 3 days of randomization. What proportion of patients is expected to require cast after 3 days? Please discuss. Sample size and planned statistical analysis.  6. The planned statistical analysis seems to be too simple. ICH E9 guidelines suggest that for between-group comparison of a continuous outcome that is measured both at baseline and follow-up, a regression adjusted for the baseline score is preferred. Further, using Shapiro-Wilk test for assessing normality of the data is not informative, as power of this test is unknown. Instead, the authors could assess the fit of chosen regression models (such as residual diagnostics for linear regression) and act thereafter. QDASH is a well-known instrument, so the authors may be able to already now know if linear model will be suitable or not. 7. Considering that QDASH will be collected at baseline, 2 weeks, 6 weeks and 3 months, a model that could make the ITT approach more feasible is a linear mixed model with all time points included, while the primary comparison made for the 3 months data. 8. When it comes to adjustment for baseline covariates in the primary model, I think that it could be a bit risky to base this on >10% change in the effect estimate. Instead, please define a priori a short set of covariates that are considered important to increase power of the model. 9. Considering all the above, I suggest reconsidering the approach to statistical analysis of the trial data. 10. The sample size would be optimally estimated taking the planned statistical model into account. Further, a range of plausible SDs could be used to assess how sensitive the sample size is with respect to the assumed variability of the data. If planning to use a mixed model (which I strongly suggest), the correlation between measurements taken from the same patient over time can naturally be considered. 11. For the mixed models planned, please mention the planned structure of random effects and estimation method (ML? REML? Method for estimation of degrees of freedom? Covariance structure?).
--	--

	12. Cross-over and handling of missing data. The authors anticipate a large drop-out (of 15%) but no methods for handling missing data are mentioned in the protocol. Further, there is no motivation for how the ITT analysis will be implemented (the analysis model as planned now will not be able to include persons lost-to-follow up). Also, the proportion of anticipated cross-overs does not seem to be taken into account. Please consider these issues as they can have a direct impact on the interpretability of the trial's results. 13. Please consider the impact of randomization stratified on centre and if it should be taken into account in the statistical analysis of the data.
--	--

VERSION 1 – AUTHOR RESPONSE

Reviewer: 1

This study is important and interesting because it questions the foundation of an established “truth” about how to treat clinically suspected scaphoid fractures, with initially normal radiographs. The fear of not immobilizing a scaphoid fracture early enough and run the risk of non-union or avascular necrosis, is most likely exaggerated. The researchers assume that most scaphoid fractures with initially normal x-rays are less likely to heal with complications.

The study design is well founded, but the following points mentioned in the attached document should be addressed:

Dear reviewer, thank you for reviewing our manuscript and your comments on our protocol. We tried to improve our protocol based on your comments and explained some choices we made for our study design below.

1. The researchers ought to be blinded for which sort of immobilisation the patient has had, when they perform their extensive examination after 2 weeks. This would make the results more trustworthy, and it is not enough to just let the radiologists be blinded. This can be done if new X-rays are taken before the researcher examines the patient. The radiographers can remove casts and bandages and make patients wash their hands prior to this examination.

Response: Dear reviewer, we understand your commend. However, we choose not to blind the assessor of the physical examination for allocated treatment, after consultation with involved departments of each participating hospital. There were multiple logistic issues because we have to deal with the clinical practice in all the participating hospitals, that made it impossible to blind the assessor. The researcher who will perform all physical examinations, does not have a preference for one of both treatment options. So, we believe that the outcome of the physical examinations will not be influenced by preferences of the assessor. For the primary outcome, QDASH at 3 months, the patient, who fills in the questionnaires, cannot be blinded. Therefore, we will perform the statistical analysis blinded for allocated treatment.

Action: Change line 209: Because of practical reasons, there is no blinding of the treatment group for the physician, patient and researcher, but assessment of the radiographs is blinded. We will perform statistical analysis blinded for allocated treatment.

2. The planned extensive examination by the researcher, 2 weeks after injury, might not be possible for patients to perform, and might even be harmful in case of a scaphoid fracture (eg measuring the hand strength with a Jamar dynamometer).

Response: The safety of the patients is our primary goal, as is part of the Good Clinical Practice. The researcher, who is a physician with clinical experience, will perform the physical examination. This

examination consists of several tests which are standardly performed in daily practice of patients with suspected scaphoid fracture. If painful, a test will not be performed.

Action: -

3. What happens with patients with too much pain to follow the short bandage randomisation group after some time? Are they excluded from the study?

Response: Thank you for your comment, we will make it more clear in our method section. We are performing intention-to-treat analysis, so all patients remain in the study and will be analyzed according to their allocated group.

Action: Change line 217 When patients experience too much pain despite adequate analgesia, an additional appointment is made at the outpatient clinic to apply a below-elbow cast for analgesic purposes. A similar below-elbow cast to the control group is applied conform hospital protocol. The patients that cross-over to cast remain in the study.

4. Is it necessary for the researcher to do the clinical examination after 2 weeks at all - unless the researchers intend to validate the physicians' clinical skills? The study would have a higher impact if the patient-flow is done as normal and close to the every-day clinical life as possible.

Response: We perform a pragmatic study. As suggested by the reviewer, the patient-flow is indeed done as normal and close to the every-day clinical life. The physician will perform their own physical examination and determines the diagnosis, treatment and follow-up without interference of the researcher according to their hospital protocol. One researcher will perform a standardized physical examination for study purposes. We chose to incorporate a clinical examination by the researcher because it is plausible there is a difference in wrist movement and grip strength between patients with a cast for 14 days and patients with 3 days supportive bandage.

Action: Add to line 124 and 132: Pragmatic.

Change line 173 Lastly, the physician re-examines the patient and determines the diagnosis, treatment and follow-up according to their hospital protocol without interference of the researcher.

5. If this physical examination after 2 weeks is done by the physician on call - as in normal practice, the researcher would be blinded for the examination after 1 year. The assessments of the different PROM-results should also be done in a blinded way.

Response: The researcher will perform a standardized physical examination on all patients after two weeks. We chose to incorporate this because it is plausible there is a difference in wrist movement and grip strength between patients with a cast for 14 days and patients with 3 days supportive bandage. The same researcher will perform the physical examination after one year. Since it is also not possible to blind patients for their treatment, we will perform data analysis blinded for treatment allocation.

Action: Change line 209: Because of practical reasons, there is no blinding of the treatment group for the physician, patient and researcher, but assessment of the radiographs is blinded. We will perform statistical analysis blinded for allocated treatment. Add line 358 We will perform all analyses blinded for treatment allocation.

6. Is it necessary to use so many different PROMS and so many follow up-periods for patients to answer the PROMS? The drop-out risk will be high because of this.

Response: Dear reviewer, we understand your concern. However, in the review process of our grant proposal, the project was also reviewed by the Dutch patient federation. No comments were made concerning the follow-up moments and the chosen questionnaires. We chose to follow standard evaluation time points as done in clinical practice, since this study has a pragmatic design. The 1-year follow-up time point is additional to this standard evaluation moment to perform a radiographic and physical examination, besides the final PROM's. Additionally, we chose the same questionnaires as used by previous studies dealing with scaphoid fractures. For the cost-effectiveness it is necessary to

have information of the medical and society related costs. Therefore at 6- and 9-months, patients have to fill in a brief questionnaire.

Action: -

7. Some PROMS address the same factors. Quick-Dash and PRWHE should be preferred and both already include a pain scale.

Response: We use the QDASH and PRWHE both, because the QDASH takes the function of both arms in account and the PRWHE takes the function for the affected arm in account. With the Quick-Dash we can reflect if the wrist problem results in an impaired functional outcome in daily life and with the PRWHE we can reflect only on the function of the affected arm. As the reviewer suggested, we indeed use the pain scale from the PRWHE to evaluate pain.

Action: -

8. Another major concern is the number of patients included. The sample size is based on significant differences in PROM scores. Even if the research question is “whether clinically suspected scaphoid fractures without radiological evidence of a fracture – have an equal outcome after 3 months if they use a bandage or a cast”, it is of some importance to address whether they have an identified scaphoid fracture or not after 2 weeks. With only 180 patients included and 10% with a scaphoid fracture (n=18), the risk of large baseline differences in the randomization is present. The risk of type 1 statistical errors is high if eg most of the fractures are immobilised in a cast.

Response: Our sample size calculation is based on the primary outcome of our study, namely the QDASH after 3 months. We chose to use a patient reported outcome measure in consultation with the Dutch patient federation and the grant supplier. We agree with the reviewer that it is interesting how many scaphoid fractures will be found in each group, and this will be reported. We believe that most of the scaphoid fractures in both groups will be united within three months and therefore will not have great influence on our primary outcome.

Action: Add line 291 (Serious) adverse events are noted, such as scaphoid fracture and scaphoid nonunion. (Detailed information can be found at safety considerations).

9. Likewise, age and gender should be equally distributed as these factors affect both pain scores and function. Should there be an upper age limit - like 50 years of age? Higher risk of radial sided pain due to CMC-osteoarthritis?

Response: Since our study has a pragmatic study design, we will include all patients with clinically suspected scaphoid fractures. Our physical examination is focused on scaphoid fractures, and not on CMC-osteoarthritis. If despite the randomization not all baseline characteristics are equally distributed, we will adjust for these imbalances if the effect estimates changes $\geq 10\%$.

Action: Add line 376 If baseline imbalances are present between the two groups, we add these as fixed effects if the effected estimate changes with more than 10%. Potential confounders we test for are: age, gender, Body Mass Index, presence of comorbidities that influence function of the arms, previous fracture of hand or wrist, diabetes, smoking status, dominant side affected, trauma mechanism (High energetic trauma, Low energetic trauma, sport, traffic accident, assault) and employment status (in school, unemployed, retired, entrepreneur, employee, occupational disabled).

10. A discussion about all the other possible types of injury in a “radial sided painful wrist after injury” is missing in the introduction. Scapholunate ligament-injuries, painful bone-bruises, etc.

Response: We understand that not all patients with radial sided wrist pain have a clinically suspected scaphoid fracture. But the focus of our study is on patients with a clinically suspected scaphoid fracture with normal initial radiographs. To explain the rationale of our study we focused the introduction on clinically suspected scaphoid fracture and treatment options. We believe that adding

other possible injuries will not clarify the aim of our study. Therefore, we suggest not to add this to our introduction. We will change radial sided wrist pain in the section about study design to clinically suspected scaphoid fracture. We will add to the method section that we report the diagnosis of our included patients as a secondary outcome.

Action: Change line 163 All patients who present with a clinically suspected scaphoid fracture after trauma at the emergency department (ED), who match the in- and exclusion criteria, are informed about the study by their treating physician.

Add line 291 (Serious) adverse events are noted, such as scaphoid fracture and scaphoid nonunion. (Detailed information can be found at safety considerations).

Other diagnosis are reported such as other carpal fractures or ligament injuries.

11. It is not discussed whether it is clinically relevant to do an X-ray of all wrists after 2 weeks. Some patients have no pain and no clinical signs at all, and the medical indication for a new X-ray might not be present.

Response: The rationale why we perform a X-ray after 2 weeks in our study is as follows. First the accuracy of an X-ray after two weeks to see a scaphoid fracture is high, especially when compared to the trauma X-ray. Because on average 10% of the fractures will be missed on the initial X-ray, and the fact we use an experimental treatment (supportive bandage), we believe that an X-ray after 2 weeks will be of additional value for our study. We agree that in clinical practice, the choice for an X-ray after 2 weeks will be driven by the clinical symptoms of the patient. Nevertheless, the perception of pain (during clinical examination) is very person dependent and missing a scaphoid fracture can have negative consequences.

Action: -

12. Sometimes an MRI or CT scan is needed after two weeks in order to identify a scaphoid fracture that is still not evident on X-rays, but still clinically suspected. These investigations are expensive and will have an impact on the cost analysis. This should also be addressed.

Response: We agree with reviewer that these expensive investigations will have impact on the cost analyses, and we will address this in our study. We will collect this information from the electronic patient status during the whole follow up period. We will make this more clear in our manuscript.

Action: Change data collection line 355 We screen the electronic patient status to collect information about (serious) adverse events, hospital visits, additional imaging and treatment for the wrist injury during the follow-up period of the study.

Add to cost analysis line 423 The use of diagnostics (additional radiographs, CT, MRI, ultrasound or bone scintigraphy), consultations, emergency department visits, physical therapy, medication and aids are evaluated.

13. A more detailed and complete flow-chart of the study is needed.

Response: We will add this to our manuscript to make it more clear.

Action: Added Figure 1. Reference in method section line 208. Add legend line 582 Figure 1 SUSPECT study flow chart. PROMs; Patient rated outcome measures

Reviewer: 2

This is an interesting study and I enjoyed reading the protocol. I have some comments below.

Dear reviewer, thank you for reviewing our manuscript and your comments on our protocol. We tried to improve our protocol based on your comments and explained some choices we made for our study design below.

In the "strengths and limitation of this study" it comments that the study is looking at supportive bandage producing equal function outcome to below-elbow cast. The design, however, is not looking at whether the two treatment options produce equal outcomes but whether the bandage is "not

unacceptably worse” than cast. The authors also state at the end of the “Introduction” that their hypothesis is that the two treatments will be equal in functional outcome. This is not the hypothesis that is being tested with the choice of trial design. Again, in the “sample size” section, it comments that looking at “minimally equal function”. I think the choice of language about being “equal” is misleading. You are willing to accept introducing into clinical practice an inferior treatment, as long as its not unacceptably worse and because of its potential cost-effectiveness. I think the design is fine, with the minimal risk to patients, but it is not producing findings that treatments are “equal”.

Response: Thank you for addressing this to us. Since it is not clear, we will change it to not inferior in the manuscript.

Action: Change equal to not inferior through the manuscript (line 97, 127, 135, 141, 338)

If <10% of patients have an occult scaphoid fracture and are concerned about over-treatment and costs, then should there be a third arm of ‘no treatment’? The occult fractures are also likely to be the lesser risk fractures to lead to non-union and, therefore, further reduces the risk of ‘no treatment’. The two-arm design is acceptable, but was a ‘no treatment’ arm considered with two week radiological follow-up and patients treated timely in a cast to prevent non-union? I understand though that it could be argued that a patient needs a little support and a bandage is low cost.

Response: We performed a pragmatic study. The standard treatment for patients with a clinically suspected scaphoid fracture is below-elbow cast. In agreement with the medical ethics committee of the Erasmus MC Medical Center, we decided to give patients supportive bandage to give them some comfort since patients will present at the emergency department with pain. We will make it more clear in our manuscript that we perform a pragmatic study.

Action: Add to line 124 and 132: Pragmatic

It should be clear in the “Study design” section what imaging (X-ray series and/or CT) the patient gets on first presentation to the hospital. The standard of imaging taken on initial presentation can help reduce missing no fracture. Even if its according to local hospital protocol, rather than an agreed trial protocol, this should be stipulated. I can understand in a pragmatic trial you may not want to be too prescriptive about what imaging is taken, but I would prefer to see at least a minimal standard of X-rays taken across the hospitals – particularly when trying not to miss a difficult fracture to spot.

Response: We have a table with all the views per hospital but we do not think it will add much to our paper. Therefore we do not add this as a table to our manuscript.

Participating hospital Number of views Scaphoid views

Albert Schweitzer Hospital, Dordrecht 5 PA with ulnar deviation, lateral wrist, 30 degrees oblique, 45 degrees oblique PA, 45 degrees oblique lateral

Haaglanden Medical Centre, The Hague 3 PA with ulnar deviation, Lateral, Supination 45 degrees

Haga Hospital, The Hague 3 PA with ulnar deviation, Supination 30 degrees, Supination 60 degrees or supination 45 degrees, Wrist PA, Wrist lateral

Erasmus MC Medical Centre, Rotterdam 5 PA with ulnar deviation, Supination 45 degrees AP, Supination 45 degrees PA, Wrist PA, Wrist Lateral

Franciscus Gasthuis and Vlietland, Rotterdam and Schiedam 4 PA wrist, lateral wrist, AP with ulnar deviation, oblique 30 degrees

Ijsselland Hospital, Rotterdam 4 PA with ulnar deviation, Lateral scaphoid, 30 degrees Supination, Oblique 45 degrees

Ikazia Hospital, Rotterdam 3 PA with ulnar deviation, Lateral, Supination 30 degrees

Maasstad Ziekenhuis, Rotterdam 3 PA with ulnar deviation, Lateral, Supination 45 degrees

Reinier de Graaf Gasthuis, Delft 4 AP with ulnar deviation, 45 degrees Supination, Wrist AP, Wrist Lateral

Action: Change line 194 No scaphoid fracture reported by the radiologist or treating physician on the initial radiographs with scaphoid specific views (minimally 3 views).

Change line 282 After 2 weeks, the radiographs include scaphoid specific views (minimally 3 views) to identify a scaphoid fracture.

Also, only two radiographs at one year (PA and lateral) to assess for non-union? This is a very limited set of imaging – if you are to bring in the patient at one year for extra little cost and radiation exposure you could collect a more thorough set of imaging and hence rigorous assessment for non-union.

Response: In our opinion, a nonunion of a scaphoid fracture appears with sclerotic changes and should therefore be visible on posteroanterior and lateral radiographs of the wrist. This pragmatic approach minimizes the radiation exposure and costs. When patients do report wrist pain after one year, we will refer them back to their treating physician and further (radiological) examination can be done.

Action: -

As would including more than one person doing this assessment.

Response: As suggested by the reviewer, the radiographs will indeed be assessed by two independent assessors. We will make this clear in the manuscript.

Action: Add line 286 The radiographs are evaluated by two independent assessors.

How is “non-union” defined?

Response: A scaphoid nonunion is defined as scaphoid fracture without signs of healing after minimally 12 weeks.

Action: Change adverse events line 466 Scaphoid non-union (scaphoid fracture without signs of healing after minimally 12 weeks).

Also, in the “Study design” section, patients will be asked to complete questionnaires at baseline and at six further follow-ups over a year. This is considerable (arguably unnecessary) patient burden, for whom over 90% won't even have a fracture. Most patients at three months will have largely recovered from their fracture, even if they have one. I would have advised against the six and nine month questionnaires, particularly when the three month questionnaire is the primary end-point, and the one year as a final check (although this final check could have been done at six months as one year on will be difficult to bring patients in for review and collect further data with no financial incentives/expenses to be covered).

Response: To report our cost-effectiveness over one year, we need these brief questionnaires at 6 and 9. All the other questionnaires will not be sent to the patients at 6 and 9 months.

Action: -

Be explicit in the “Intervention group” about how the below-elbow cast will be applied like it is for the “control group”.

Response: We added information how the cast will be applied.

Action: Change line 217 When patients experience too much pain despite adequate analgesia, an additional appointment is made at the outpatient clinic to apply a below-elbow cast for analgesic purposes. A similar below-elbow cast to the control group is applied conform hospital protocol. The patients that cross-over to cast remain in the study.

Also, how did you arrive at “3 days”? You could have a lot of patients coming in for cast because still in pain after only 3 days – this could lead to a lot of cross-over?

Response: In daily practice joint sprains are treated with a supportive bandage for approximately 3 days to decrease pain and swelling. Based on this we chose 3 days. When patients do cross over, they will remain in the study since we will perform intention-to-treat analysis. We will report the cross-

over, because this is also an important outcome in our study. We will make it more clear in our manuscript that we will report the amount of cross-over.

Action: Add figure 1.

Add line 287 Detailed information about the received treatment is reported. We address the number of patients that cross over from their allocated treatment , days of cast or supportive bandage before the outpatient department appointment and applied treatment after the outpatient department appointment (cast or surgery).

I am not sure what the protocol is when a fracture is found at the two week review. When a patient does have a fracture and its only being treated in cast how are they still followed-up to monitor for non-union? If non-union does occur will surgery be performed? If in the bandage group and there is a fracture after the two week review, do you switch to cast or keep with the bandage?

Response: We are performing a pragmatic study. We will not interfere with hospital protocol according to treatment or follow up for scaphoid fracture, scaphoid non-union or other post-traumatic injuries. When the treating physician reports a scaphoid fracture, patients can be treated with cast or surgery. This is independent of the group allocation and study protocol. We will make this more clear in our manuscript.

Action: Add to line 124 and 132: Pragmatic.

Add line 175 When a scaphoid fracture is diagnosed by the physician, patients are treated for a scaphoid fracture according hospital protocol with either cast or surgery.

In the “sample size” section it is specified that the margin of non-inferiority is 7.50 points on the QDASH. The standard deviation after three months is expected to be 14 points. This equates to a large effect size of 0.53. This appears to mean that you are willing to accept the bandage being up to half a standard deviation inferior on the QDASH. Is this an acceptable amount of inferiority?

Response: The choice of a clinically acceptable difference as margin of non-inferiority is arbitrary and should be smaller than the clinically relevant difference of the outcome score. We chose in collaboration with our statistician 50% of the margin of clinical minimal important difference of QDASH. We believe that this margin is acceptable to evaluate whether a supportive bandage is not-inferior compared to a below-elbow cast.

Action: -

Reviewer: 3

Dear reviewer, thank you for reviewing our manuscript and your comments on our protocol. We tried to improve our protocol based on your comments.

Line 132-135 Objectives: the authors state “The primary objective of this study is to determine if treatment with supportive bandage compared with below-elbow cast in patients with clinically suspected scaphoid fracture and normal initial radiographs, results in equal functional outcome after 3 months” – however the authors are conducting a non-inferiority trial, not an equivalence trial. Can the authors clarify here that the objective is to test that supportive bandage is not inferior to below elbow cast– rather than testing for equivalence, which i assume it is given the design.

Response: Thank you for addressing this to us. Since it is not clear, we will change it to not inferior in the manuscript.

Action: Change equal to not inferior through the manuscript (line 97, 127, 135, 141, 338)

Line 138-140 “we assess post-traumatic injuries, pain, patient satisfaction of the received treatment, functional outcome measured with the QDASH, recovery of function measured with physical examination and the Patient Rated Wrist/Hand 140 Evaluation (PRWHE) and quality of life ”Are secondary outcomes being assessed for non-inferiority (as the primary objective) or will be superiority be assessed for these? Please further clarify the secondary objectives.

Response: Our hypothesis is that patients with a clinically suspected scaphoid fracture without a fracture on initial radiographs treated with a bandage, have not inferior functional outcome after 3 months compared to cast, but with lower costs. So, the cost-effectiveness will be assessed for superiority. All other secondary outcomes will be assessed for non-inferiority. We will make this more clear in our manuscript.

Action: Changes secondary objectives line 141 Furthermore, we assess if supportive bandage compared to cast results in not inferior pain, patient satisfaction of the received treatment, functional outcome measured with the QDASH, recovery of function measured with physical examination and the Patient Rated Wrist/Hand Evaluation (PRWHE), and quality of life. Add to data analyse line 373 All outcomes are assessed for non-inferiority, expect cost-effectiveness which is assessed for superiority.

Line 149 – Patient and public involvement statement – Great that the authors has a patient panel.

Can the authors add how many patients made up the patient panel?

Response: We asked 3 patients to be part of our patient panel.

Action: Change line 153 In the absence of a patient association, we formed a panel of 3 patients with a (suspected) scaphoid fracture to think along with, and comment on our study.

Line 197 Randomization – presumably the computer-based randomization enables allocation concealment i.e. although an open trial those involved in recruitment are not aware of the next randomisation? Can the authors explicitly state this if this is the case for transparency on allocation concealment.

Response: The treating physician will perform the randomization at the emergency department through computer-based variable block randomization by Castor Electronic Data Capture. They have to login to a website (central randomization) and create a new patient. Since it is a variable block randomization, the physician is not aware of the next randomization. (line 201-202)

Action: Change line 204 After written informed consent is obtained, patients are central randomized through computer-based variable block randomization (2, 4, 6 blocks) by Castor Electronic Data Capture (EDC) to enable allocation concealment,(13).

Sample size – line 325 – can the authors clarify the primary hypothesis of the study. In a non-inferiority trial, the null hypothesis is that the new treatment is inferior to the standard control treatment. The alternative research hypothesis is that the experimental treatment is not inferior to the standard control arm by the pre-specified margin.

Response: Thank you for addressing this to us. Since it is not clear, we will change it to not inferior in the manuscript.

Action: Change equal to not inferior through the manuscript (line 97, 127, 135, 141, 338)

Line 354 - It is written that "Adjustment are done for those baseline variables that change the effect estimate with more than 10%" This is an usual approach to take. Can the authors provide the list of potential prognostic variables that will be assessed using this approach? it is recommend that only variables known a priori to be strongly/moderately related with the primary outcome be considered as covariates and pre-specified In the protocol and that only a few justified covariates should be included in a primary analysis. (see https://www.ema.europa.eu/en/documents/scientific-guideline/guideline-adjustment-baseline-covariates-clinical-trials_en.pdf which states "Known or expected associations with the primary outcome variable should be justified on the basis of previous evidence (e.g. data from previous trials) and/or on clinical grounds. The reasons for including a covariate in the primary analysis should be explicitly stated in the protocol.")

Response: Based on the results of previous studies we will include baseline variables as potential covariates in our mixed model.

Action: Add line 376 If baseline imbalances are present between the two groups, we add these as fixed effects if the effected estimate changes with more than 10%. Potential confounders we test for are: age, gender, Body Mass Index, presence of comorbidities that influence function of the arms, previous fracture of hand or wrist, diabetes, smoking status, dominant side affected, trauma mechanism(High energetic trauma, Low energetic trauma, sport, traffic accident, assault) and employment status (in school, unemployed, retired, entrepreneur, employee, occupational disabled).

Will the baseline outcome be adjusted for within the analysis? This should usually be included as a covariate. (see https://www.ema.europa.eu/en/documents/scientific-guideline/guideline-adjustment-baseline-covariates-clinical-trials_en.pdf)

Response: According to the statistician we consulted, our mixed model is more stable when we include the baseline outcome as one of the repeated measures instead as a covariate.

Action: Add line 372 Baseline, and 2 weeks, 6 weeks, and 3 months follow up are included in our model as repeated measures.

Add line 389 Furthermore, we assess the progress of the QDASH, PRWHE, VAS, EQ5D-5L and physical examination during the 1 year follow-up. The random and fixed effects will remain similar, only all follow up moments will be included in the repeated measurements.

Given randomisation is stratified by site I was surprised that it is not planned to adjust for site in the analysis (see above reference and Kahan, B.C. and Morris, T.P. (2013), Analysis of multicentre trials with continuous outcomes: when and how should we account for centre effects?. *Statist. Med.*, 32: 1136-1149. doi:10.1002/sim.5667 and Kahan, B.C. and Morris, T.P. (2012), Improper analysis of trials randomised using stratified blocks or minimisation. *Statist. Med.*, 31: 328-340. doi:10.1002/sim.4431). It is recommend that analysis should adjust for recruitment centre to obtain the appropriate standard errors.

Response: We agree with the reviewer and we will add hospital as a random intercept.

Action: Add line 370 Hospital is a random intercept and to analyze repeated measurements, we add a random intercept for patients (nested with hospital).

No details on provided how missing data will be handled. Given up to 15% anticipated in sample size calculations please provide details on how missing data will be handled in the primary analysis. Are any sensitivity analyses planned if there is up to 15% missing data? methods for sensitivity analysis should be included where planned.

Response: We will provide more details in our manuscript.

Action: Add line 384 Patients are included in our mixed models when they responded to the outcome questionnaire at either 2 weeks ,6 weeks or 3 months follow up. When patients respond to neither of these measurements periods, the patient will be scored as loss to follow up. When one or more (but not all) of the measurements are missing, a sensitivity analysis will be performed to clarify if the missing data is at random.

Line 356-60: Secondary scores will be analysed with a linear mixed models. The authors list the fixed effects – presumably treatment should also be included here? Will the trialists be assessing the treatment effect over all time periods, i.e. analysis model will include just a single fixed treatment effect? Or will the model include a treatment by time interaction to assess the treatment effect at each time point separately?

Response: The model will indeed include a treatment by time interaction so we can assess the effect of treatment at each time point.

Action: Add line 369 For our primary outcome, QDASH at 3 months, and secondary outcomes PRWHE, VAS score, EQ5D-5L at measurement time, and an interaction term of measurement time by treatment.

Can the authors indicate whether there will be any random effects in the mixed model? And if so which ones?

Response: We will include hospital as a random effect in our model because we stratified for hospital. Since we analyze repeated measurements, we included patients also as a random intercept (nested with hospital)

Action: Add line 370 Hospital is a random intercept and to analyze repeated measurements, we add a random intercept for patients (nested with hospital).

Are secondary outcomes being assessed for non-inferiority or will a superiority/other comparison be assessed for these?

Response: Our hypothesis is that patients with a clinically suspected scaphoid fracture without a fracture on initial radiographs treated with a bandage, have not inferior functional outcome after 3 months compared to cast, but with lower costs. So, the cost-effectiveness will be assessed for superiority. All other secondary outcomes will be assessed for non-inferiority. We will make this more clear in our manuscript

Action: Changes secondary objectives line 141 Furthermore, we assess if supportive bandage compared to cast results in not inferior pain, patient satisfaction of the received treatment, functional outcome measured with the QDASH, recovery of function measured with physical examination and the Patient Rated Wrist/Hand Evaluation (PRWHE), and quality of life.

Add to data analyse line 374 All outcomes are assessed for non-inferiority, expect cost-effectiveness which is assessed for superiority.

How will adverse events be described/analysed? and for what analysis population? Please provide details.

Response: We will report the number of adverse events per allocated treatment group.

Action: Add to secondary outcome line 291 (Serious) adverse events are noted, such as scaphoid fracture and scaphoid nonunion. (Detailed information can be found at safety considerations).

Add to data analysis line 363 We report the proportion of diagnosis and adverse events per allocated treatment group.

Cost effectiveness analysis - please provide details on how missing data will be handled within these analysis.

Response: We will provide more details in our manuscript

Action: Add line 412 We impute missing data by multiple imputation, using the pattern mixture model.

Can the authors include a brief update on trial status/dates, e.g.. On what date did recruitment begin and when is data for last recruitment/follow-up anticipated?

Response: During submission of our study protocol we were still recruiting patients. We started recruitment on 12-06-2018. The last follow-up visit will be one year after the last patients is included.

Action: -

Reviewer: 4

The protocol describes details of a randomized multicentre study to evaluate if supportive bandage is non-inferior to below-elbow cast in patients with a clinically suspected scaphoid fracture. The primary outcome is QDASH measured at 3 months after randomization and the study is performed in 9 hospitals in Netherlands. The planned non-inferiority margin is half of clinically important difference.

Dear reviewer, thank you for reviewing our manuscript and your comments on our protocol. We improved our manuscript based on your comments and explained some choices we made for our study design below.

1. The study assumes block randomization stratified by centre with varying block size, with allocation ratio 1:1. Could you please clarify how the randomization is done in practice? Does the treating physician obtain the treatment allocation through the hospital computer system? Or will numbered sealed envelopes be used?

Response: Thank you for your comment, we will make it more clear in our manuscript. The treating physician will perform the randomization at the emergency department through computer-based variable block randomization by Castor Electronic Data Capture. They have to login to a website (central randomization) and create a new patient. Since it is a variable block randomization, the physician is not aware of the next randomization.

Action: Change line 204 After written informed consent is obtained, patients are central randomized through computer-based variable block randomization (2, 4, 6 blocks) by Castor Electronic Data Capture (EDC) to enable allocation concealment,(13).

2. Will the proportion of patients required below-elbow cast at the 14 days visit be reported? If it would theoretically be that most of the patients still require cast (even with equal functional results at 3 months) that would suggest a potential need not to delay the cast treatment. Could you please comment on this?

Response: We agree with the reviewer that this is important to report, since the duration of treatment is also an important outcome. We will elucidate this in our manuscript.

Action: Add to secondary outcomes line 287 Detailed information about the received treatment is reported. We address the number of patients that cross over from their allocated treatment , days of cast or supportive bandage before the outpatient department appointment and applied treatment after the outpatient department appointment (cast or surgery).

3. It would be preferable if the examining physician for 1-year outcome was not aware of the treatment received. Is this feasible?

Response: We chose to incorporate a clinical examination by the researcher at 14 days because it is plausible there is a difference in wrist movement and grip strength between patients with a cast for 14 days and patients with 3 days supportive bandage. The same researcher will perform the physical examination after one year. Because the researcher is not blinded at 14 days, the researcher is aware of the treatment.

Action: -

4. Even if blinding of treating physicians is not possible, it should be possible for the investigator and the statistician to be blinded. Please clarify if this is planned or not and why.

Response: We perform a pragmatic study, so the patient-flow is done as normal and close to the every-day clinical life. The physician will perform their own physical examination and determines the diagnosis, treatment and follow-up without interference of the researcher according to their hospital protocol. One researcher will perform a standardized physical examination for study purposes. For the primary outcome, QDASH at 3 months, the patient, who fills in the questionnaires, cannot be blinded. For the secondary outcomes, the assessment of the radiographs can be blinded for allocated treatment. Concerning the physical examination; we choose not to blind the assessor of the physical examination for allocated treatment, after consultation with involved departments of each participating hospital. There were multiple logistic issues because we have to deal with the clinical practice in all the participating hospitals, that made it impossible to blind the assessor. The researcher who will perform all physical examinations, does not have a preference for one of both treatment options. So, we believe that the outcome of the physical examinations will not be influenced by preferences of the assessor. We will perform the statistical analysis blinded for allocated treatment.

Action: Change line 209: Because of practical reasons, there is no blinding of the treatment group for the physician, patient and researcher, but assessment of the radiographs is blinded. We will perform statistical analysis blinded for allocated treatment. Add line 358 We will perform all analyses blinded for treatment allocation.

5. The study design includes a possibility of cross-over from supportive bandage group to the cast group, when cast is needed for analgesic purposes after 3 days of randomization. What proportion of patients is expected to require cast after 3 days? Please discuss.

Response: We expect maximum 10% cross-over. We will analyze the patients in their original allocated group, since we are performing intention-to-treat analysis.

Action: -

6. The planned statistical analysis seems to be too simple. ICH E9 guidelines suggest that for between-group comparison of a continuous outcome that is measured both at baseline and follow-up, a regression adjusted for the baseline score is preferred. Further, using Shapiro-Wilk test for assessing normality of the data is not informative, as power of this test is unknown. Instead, the authors could assess the fit of chosen regression models (such as residual diagnostics for linear regression) and act thereafter. QDASH is a well-known instrument, so the authors may be able to already now know if linear model will be suitable or not.

Response: In previous studies with the QDASH as a primary outcome, linear models are used.

Therefore, we think a linear model will be feasible for us. However, we will test the assumptions of our model and check if the linearity assumptions are met.

Action: Add line 360 The distribution analysis of baseline variables is tested by the Shapiro-Wilk test. For normally distributed variables, parametric tests are used. For those variables that are not normally distributed, non-parametric tests are used.

Add line 364 All outcomes are analyzed 'by intention to treat analysis' with a linear mixed model with specified fixed and random effects. The covariance structure is unstructured. Restricted maximum likelihood will be used to estimate parameters. The assumptions of our model (linearity, homoscedasticity and normal distribution of the error terms) are tested. Should any of these assumptions seriously fail, variable transformations are used.

7. Considering that QDASH will be collected at baseline, 2 weeks, 6 weeks and 3 months, a model that could make the ITT approach more feasible is a linear mixed model with all time points included, while the primary comparison made for the 3 months data.

Response: We agree with the reviewer that we should add these time points.

Action: Add line 372 Baseline, and 2 weeks, 6 weeks, and 3 months follow up are included in our model as repeated measures. We perform post-hoc analyses to compare estimated means at 3 months.

8. When it comes to adjustment for baseline covariates in the primary model, I think that it could be a bit risky to base this on >10% change in the effect estimate. Instead, please define a priori a short set of covariates that are considered important to increase power of the model.

Response: Based on the results of previous studies we will include baseline variables as potential covariates in our mixed model.

Action: Add line 376 If baseline imbalances are present between the two groups, we add these as fixed effects if the effected estimate changes with more than 10%. Potential confounders we test for are: age, gender, Body Mass Index, presence of comorbidities that influence function of the arms, previous fracture of hand or wrist, diabetes, smoking status, dominant side affected, trauma mechanism(High energetic trauma, Low energetic trauma, sport, traffic accident, assault) and employment status (in school, unemployed, retired, entrepreneur, employee, occupational disabled).

Considering all the above, I suggest reconsidering the approach to statistical analysis of the trial data.

Response: Thank you for your suggestions. We hope our analysis are now clearer.

Action:

10. The sample size would be optimally estimated taking the planned statistical model into account. Further, a range of plausible SDs could be used to assess how sensitive the sample size is with respect to the assumed variability of the data. If planning to use a mixed model (which I strongly suggest), the correlation between measurements taken from the same patient over time can naturally be considered.

Response: Thanks for your remark and we agree that a mixed model will be a more appropriate statistical test.

Action: Add line 364 All outcomes are analyzed 'by intention to treat analysis' with a linear mixed model with specified fixed and random effects

11. For the mixed models planned, please mention the planned structure of random effects and estimation method (ML? REML? Method for estimation of degrees of freedom? Covariance structure?).

Response: We will not make any assumption on the covariance structure and random effect structure (unstructured). Parameters will be estimated by REML. For analysis we will use the R package nmls. We are not aware which method the package uses to estimate the degrees of freedom.

Action: Add line 365 The covariance structure is unstructured. Restricted maximum likelihood will be used to estimate parameters.

12. Cross-over and handling of missing data. The authors anticipate a large drop-out (of 15%) but no methods for handling missing data are mentioned in the protocol. Further, there is no motivation for how the ITT analysis will be implemented (the analysis model as planned now will not be able to include persons lost-to-follow up). Also, the proportion of anticipated cross-overs does not seem to be taken into account. Please consider these issues as they can have a direct impact on the interpretability of the trial's results.

Response: We will provide more details in our manuscript. Additionally we will also perform as-treated analysis.

Action: Add line 384 Patients are included in our mixed models when they responded to the outcome questionnaire at either 2 weeks, 6 weeks or 3 months follow up. When patients respond to neither of these measurement periods, the patient will be scored as loss to follow up. When one or more (but not all) of the measurements are missing, a sensitivity analysis will be performed to clarify if the missing data is at random.

Add line 381 We perform additional as-treated analysis to analyze outcomes between patients that received only supportive bandage until the outpatient department appointment after 2 weeks and patients that received cast in the first 2 weeks (received cast at the ED or crossover from bandage).

13. Please consider the impact of randomization stratified on centre and if it should be taken into account in the statistical analysis of the data.

Response: We will include hospital as a random effect in our model because we stratified for hospital. Since we analyze repeated measurements, we included patients also as a random intercept (nested with hospital)

Action: Add line 370 Hospital is a random intercept and to analyze repeated measurements, we add a random intercept for patients (nested with hospital).

VERSION 2 – REVIEW

REVIEWER	Stephen Brealey University of York, United Kingdom
REVIEW RETURNED	22-Jun-2020

GENERAL COMMENTS	I am satisfied with the authors response to the comments I made about the manuscript.
---

REVIEWER	Suzie Cro Imperial College London, UK.
REVIEW RETURNED	04-Jul-2020

GENERAL COMMENTS	The authors have adequately addressed my previous comments.
---

REVIEWER	Aleksandra Turkiewicz Lund University, Sweden
REVIEW RETURNED	07-Jul-2020

GENERAL COMMENTS	Thank you for all the clarifications made and in particular for changing the statistical analysis approach to mixed models. There are still a few issues remaining.  1. I still think that testing for post-randomization imbalance is not a good idea (in line with one other reviewer). Randomization per definition leads to lack of systematic differences. If there are few important prognostic variables (such as age and sex), please plan adjusting for them already now, irrespectively of the “imbalance”. At the moment, the authors plan to test a large number of variables in an unspecified way which can induce a lot of “forking paths” problems and does not seem to have any benefits. Please reconsider. 2. Missing data. What does it mean that “a patient will be scored as lost to follow-up”? For the ITT analysis, all randomized patients need to be included in the model. Considering that a mixed model will be used, it is enough that a patient has one measurement (for example baseline) to be included in the model. Is it expected that all randomized patients will provide baseline data? If not, how will be the ITT principle implemented for persons with all data missing (an alternative is a so called full analysis set, please see EMEAs ICH Topic E 9 Statistical Principles for Clinical Trials)? Please clarify. 3. Further, from the answers to the reviewers’ comments it seems that there is possibility of missing data due to pain and not performing a task. Does this apply to the primary outcome? How will this missingness (that depends among others on the pain status) be treated? Is it possible to expect that the pain variable will not be missing and can be used to impute the other missing data? Or how will this be handled? Please clarify.
---

VERSION 2 – AUTHOR RESPONSE

A point by point feedback:

Reviewer: 4

Thank you for all the clarifications made and in particular for changing the statistical analysis approach to mixed models.

Dear reviewer, thank you for addressing the remaining issues to us. We improved our manuscript based on your following comments.

1. I still think that testing for post-randomization imbalance is not a good idea (in line with one other reviewer). Randomization per definition leads to lack of systematic differences. If there are few important prognostic variables (such as age and sex), please plan adjusting for them already now, irrespectively of the “imbalance”. At the moment, the authors plan to test a large number of variables in an unspecified way which can induce a lot of “forking paths” problems and does not seem to have any benefits. Please reconsider.

Response: We agree with the reviewer that “randomization per definition” leads to lack of systemic differences. But because of the relatively small number of patients included in our study, some variables can be unbalanced despite a correct randomization. To overcome this we changed this in the statistical analysis paragraph.

Action: Change line 374 Baseline characteristics between the control and intervention group can be unbalanced because our groups are relatively small. Therefore, we add the potential confounders age, gender, and presence of comorbidities that influence the function of the arms, as fixed effects to our model.

2. Missing data. What does it mean that “a patient will be scored as lost to follow-up”? For the ITT analysis, all randomized patients need to be included in the model. Considering that a mixed model will be used, it is enough that a patient has one measurement (for example baseline) to be included in the model. Is it expected that all randomized patients will provide baseline data? If not, how will be the ITT principle implemented for persons with all data missing (an alternative is a so called full analysis set, please see EMEAs ICH Topic E 9 Statistical Principles for Clinical Trials)? Please clarify.

Response: We understand that our description of the ITT analysis is not transparent enough. Patients that responded to the outcome questionnaire irrespectively of the number of time points will be included in the model. We expect that missing data will be random. This because we expect that the availability of data will not be related to the allocation, since patients are allowed to crossover. When there are patients without available data at all, we will report this in detail in our manuscript and discuss the potential bias in the discussion paragraph.

Action: Change line 385 Patients are included in our mixed models when they responded to the outcome questionnaire irrespectively of the number of time points. When randomized patients do not respond to the outcome questionnaire at one of the time points, we report the reason of no availability of data in detail in our manuscript and discuss the potential bias.

3. Further, from the answers to the reviewers’ comments it seems that there is possibility of missing data due to pain and not performing a task. Does this apply to the primary outcome? How will this missingness (that depends among others on the pain status) be treated? Is it possible to expect that the pain variable will not be missing and can be used to impute the other missing data? Or how will this be handled? Please clarify.

Response: When patients are in too much pain, the grip strength cannot be tested. This is not our primary outcome. Grip strength is measured with a hand dynamometer. If patients are unable to perform this test because of the pain, their grip strength will be scored as 0 kilogram.

Action: -